# Plant Use Adaptation in Pamir: Sarikoli Foraging in the Wakhan Area, Northern Pakistan

**DOI:** 10.3390/biology11101543

**Published:** 2022-10-21

**Authors:** Muhammad Abdul Aziz, Zahid Ullah, Muhammad Adnan, Renata Sõukand, Andrea Pieroni

**Affiliations:** 1Department of Environmental Sciences, Informatics and Statistics, Ca’ Foscari University of Venice, Via Torino 155, 30172 Venezia, Veneto, Italy; 2University of Gastronomic Sciences, Piazza Vittorio Emanuele II 9, 12042 Pollenzo, Bra, Italy; 3Center for Plant Sciences and Biodiversity, University of Swat, Kanju 19201, Pakistan; 4Department of Botanical and Environmental Sciences, Kohat University of Science and Technology, Kohat 26000, Khyber Pakhtunkhwa, Pakistan; 5Department of Medical Analysis, Tishk International University, Erbil 4401, Kurdistan, Iraq

**Keywords:** Broghil, Sarikoli, Wakhi, wild food plants, traditional food, cross border trade, Wakhan

## Abstract

**Simple Summary:**

In this cross-cultural ethnobotanical study, the food uses of wild food plants (WFPs) were recorded among two linguistic groups, i.e., the Sarikoli and the Wakhi people, living in the Broghil Valley of North Pakistan. A total of 29 wild food taxa were recorded. We found that the local knowledge linked to wild food plants has been completely homogenized, indicating that the Sarikoli, a cultural diaspora, have gradually adopted the traditional food system of the Wakhi. Moreover, the local knowledge of WFPs is gradually eroding in the two considered communities due to certain socio-political factors. In particular, legal sanctions on accessing natural resources and cross-border movement at the Pakistan–Afghan border are major issues that hinder socio-cultural sustainability among the two groups. Therefore, it is recommended that the legal restrictions be properly revised, and an inclusive policy should be adopted in this regard. In addition, local knowledge on food resources should be reinforced in future development and educational programs.

**Abstract:**

The study recorded the food uses of wild food plants (WFPs) among the Sarikoli diaspora and the dominant Wakhi in Broghil Valley, North Pakistan, to understand their food adaptation, mainly by looking through the lens of food ethnobotanies. A total of 30 participants took part in the study, which included 15 elderly individuals from each ethnic group. Data were gathered through semi-structured interviews. We recorded 29 WFPs, mostly used as cooked vegetables and snacks. The food uses, as well as the local plant nomenclatures, linked to WFPs of the two studied groups were completely homogenized, which could be attributed to the cultural assimilation of the Sarikoli people to Wakhi culture. We found that although traditional knowledge on WFPs has been homogenized, social change in nearby regions is also threatening the traditional knowledge of the two communities, as evidenced by the smaller number of plants reported compared to that of all other field ethnobotanical studies conducted in nearby regions. Moreover, the growth of legal restrictions and sanctions on accessing natural resources are posing serious challenges to cultural resilience in the valley, and the restrictions on cross-border movement in particular are creating challenges for those who have cross-border kinship relationships between the two groups. We suggest specific measures, such as the promotion of food tourism and educational activities, to protect traditional knowledge and bicultural heritage from further erosion in the region.

## 1. Introduction

Cross-cultural adaptation connected to transmigration is a well-known phenomenon, mainly experienced by small groups of migrants or diasporas that immerse themselves in multicultural societies and go through intercultural transitions [1,2]. According to a United Nations Report, in 2015, almost 244 million people lived outside of their home countries or territories [3]. Migrant societies usually arrive along with a wide array of cultural identities and distinct social behaviors, and most importantly they bring local/traditional practices which make them distinctive. The close interconnections among the various groups create a backdrop of socio-cultural negotiations, as cultural boundaries are not very rigid, but rather elastic, stretching and forming a state of transition, and thus cross-cultural settings create a blend of practices where small ethnic groups gradually acculturate themselves to the dominant cultures [4]. On the one hand, intercultural adaptation processes lead to social integration, but on the other hand they challenge the social identity of a given ethnic group, which consequently comes at the expense of the survival of its biocultural heritage, ultimately depleting biocultural diversity [5].

Plant-centered local knowledge is an important part of biocultural heritage and plays an important role in sustaining human life. Wild food plants (WFPs) have remained an important ingredient of the traditional food basket since pre-Neolithic times. Historically, local and traditional food systems have given sufficient space to WFPs, and their existence in daily food practices among local communities could be a parameter to qualitatively measure the socio-cultural histories, long-term human-ecological practices, and the economic instabilities of different communities on a spatiotemporal basis [6]. Therefore, ethnobotany could play an important role in describing communal changes over time and space, helping relevant stakeholders formulate future developmental programs to achieve social and environmental sustainability, especially among communities living in mountainous and underprivileged areas.

Broghil Valley, which is located at the extreme north of District Chitral, Pakistan, is home to the Wakhi and Sarikoli people. The Wakhi are the dominant ethnic community across the Pamir region and have inhabited the area for centuries. The Sarikoli diaspora originally belonged to the contiguous Taxkorgan County in Xinjiang, China [7], but migrated to the area by the mid-20th century, as confirmed by members of the local community during our research study. Even though the Sarikoli have a very different historical and cultural stratification, they are now culturally homogenized with the dominant Wakhi culture in the valley. It is interesting to note that in the study area they maintain their social representation and identity as “*Sarikoli people*”.

Traditional ecological knowledge (TEK) around food and natural resources is an important part of the biocultural heritage of the Wakhi and Sarikoli. Thus far, these groups have not been studied in terms of their food related ecological practices. Moreover, systematic research on food-centered practices is very limited throughout the whole mountainous belt of the Pamir and Hindukush mountain ranges. A literature survey revealed very few food ethnobotanical studies that have been carried out in these regions. On the Pakistani side of the Hindukush range, our own research group has recently investigated the use of WFPs in Chitral and Gilgit-Baltistan [8,9,10]. Only a single study was found on the ethnobotany of the Kyrgyz and Wakhi in Afghan Pamir [11]. Moreover, we found some ethnobotanical studies which recorded medicinal plant uses from the Tajik and Afghan Pamirs [12,13,14]. In Tajikistan, our research group has conducted a wild food ethnobotanical study in the Varzob Valley [15]. Similarly, the ethnobotanical study of medicinal teas from Southern Xinjiang, China is an important contribution to the field of the ethnobotany of mountain regions [16]. However, keeping in mind the vast geography and cultural richness near the Pamir region, or “roof of the world”, there is a dire need to explore the food diversity and food-centered local ecological practices of this area in future scientific research. It is worth mentioning that mountain areas are always considered sites of “*biocultural refugia*” [17,18], and therefore systemic overviews of local food practices among mountainous communities are essential. The current study will play an important role in understanding WFP-centered local ecological practices, which will also fill gaps in the food ethnobotanical literature.

The present research will record WFP uses among the Wakhi and Sarikoli peoples, who underwent remarkable sociocultural negotiations in the past century. The research will help to understand the similarities and differences in the food ethnobotanies of these two ethnic groups, which can be used as a proxy to determine the impacts of the dominant Wakhi culture on the Sarikoli cultural diaspora, which will have important implications for further scientific discoveries.

The specific objectives of the study were to: (a) assess the wild food ethnobotany of the considered area; (b) compare the gathered data among Sarikoli and Wakhi communities in order to understand the historical cultural adaptation processes that these groups underwent, especially that of Sarikoli speakers to the dominant Wakhi culture; and (c) compare the recorded data with previous ethnobotanical data of adjacent areas within and across the border, especially representing Wakhi ethnobotany, in order to understand the cultural assimilation processes experienced by the Sarikoli in the study area.

## 2. Materials and Methods

### 2.1. Study Area and Communities

The study area, Broghil Valley, is located in the extreme north of District Chitral, North Pakistan (Figure 1). This remote area is part of Pakistani Wakhan, which shares its borders with Afghan Wakhan, and is also contiguous with the border of China and Tajikistan (Figure 1). The area is part of the Hindukush Mountain range and its elevation ranges from 3280 m.a.s.l. (meters above sea level) in the village of Kishmanjah to 4304 m.a.s.l. at Karambar Lake in the northeast. The terrain is undulating with mountains, grassy plains, and valleys. It includes almost 3400 ha of peatland and lakes. The valley was declared a national park in 2010 (see here: https://thehighasia.com/broghil-national-park-in-naya-kp-irregularities-in-appointments-angers-locals/#:~:text=The%20whole%20Broghil%20Valley%20was,is%20also%20situated%20in%20Broghil, accessed on 10 October 2022).

The area is home to around 200 families, and mainly populated by the Wakhi people, who actually derive from Afghan Wakhan. In the valley, in the last village, known as Lashkargaz, there are also 60 Sarikoli individuals who are part of the population. Sarikolis are an ethnic diaspora living in Broghil Valley. Literature on their ethnography is very scarce, but some sources [7] have revealed that they were originally inhabitants of Western Xinjiang, China, and they are known to live in the north and northwest of the Himalayan Mountain range, as well as in nearby parts of Afghanistan and Tajikistan, especially the higher valleys, approximaly 3000 m.a.s.l. in the eastern Pamir Mountains. More specifically, the Sarikoli people originally came from Taxkorgan Tajik Autonomous County, which is an autonomous county of Kashgar Prefecture in Western Xinjiang. The county seat is the town of Tashkurgan and it represents the only Tajik autonomous county in China. It is situated at an altitude of 3090 m.a.s.l. (10,140 ft) on the borders of both Afghanistan and Tajikistan, and close to the borders of Kyrgyzstan and Pakistan. The majority of the population in the town are ethnic Mountain Tajiks. Most of the people in the region speak Sarikoli [19], which is an eastern Iranian language, and they are distinct from their Turkic neighbors. An older member (a 63-year-old man) of the Sarikoli community in Broghil Valley described their migration story in these words: “for the first time, our grandfather migrated to the area in 1932 from Taxkorgan, Xinjiang, but after a few years, in 1942, he went back to Taxkorgan. The last migration to Broghil took place in 1948 after the arrival of the Communist political system. Since our grandfather was a traditional healer, he fled and came with his two sons and settled in this area. Now we are the Sarikoli, but we have lost our language although we have relatives across the border in China. Now we have marriages with the Wakhi and Kho peoples”. According to one estimate, there are 40 thousand Sarikoli speakers spread over the adjoining areas of the abovementioned countries [19]. It is worth mentioning that Sarikoli is the only Indo-European language spoken in China, and it is one of the more poorly described languages due to the group’s physical and political marginalization from other speakers of Pamir languages [19]. In the area, the Sarikoli are closely surrounded by Wakhi speakers and the Kho, who are also their close neighbors in the lower valleys of District Chitral. In Broghil Valley, there are only 60 Sarikoli individuals, while Sarikolis (18,000 population) exceed Wakhis (1600 population) in Taxkorgan County of China [20]. 

Both of the studied communities practice pastoralism and small-scale horticulture. They mainly grow barley and wild beans to supplement their traditional food system, which primarily consists of meat, dairy products, and rice, the latter of which is imported from the nearby local food market.

### 2.2. Ethnobotanical Survey

A field ethnobotanical survey was carried out in the month of June 2021, in different villages in Broghil Valley. The study aimed to record WFP uses among the two researched groups. A total of 30 people, 15 informants from each study group (Table 1), were interviewed (Figure 2).

The participants were mainly mature individuals, more than 45 years of age, who had long-term experience with nature and the local environment. The sample size was small because the Sarikoli are a diaspora represented by a total of 60 individuals, including children and young people, and therefore we were able to select only 15 individuals among them. In order to balance the number of individuals for both sample sets, we selected 15 individuals among the Wakhi as well (Table 1). Prior to each interview, we received verbal consent to share their knowledge and take photos for the completion of the study. Most of the participants shared their ethnobotanical information on WFPs in their local languages which were translated into Urdu by a local guide and translator. We strictly followed the recommendations provided by the International Society of Ethnobiology [21]. The study was part of the PhD research work conducted by the first author and was approved by the University Ethics Committee of the University of Gastronomic Sciences Pollenzo, Italy. The interviews mainly focused on gathered WFPs, i.e., plants used as cooked vegetables, snacks, recreational teas, and seasoning. We also explored the uses of WFPs in lacto-fermentation processes and dairy products. Each plant taxon was recorded along with the local name, which was revised many times to avoid any error and discrepancy in local nomenclature. Qualitative ethnographic data was obtained through direct observations and in some cases through open-ended questions. At the end of the survey, recorded plant taxa were gathered, and the collected specimens were identified by the third author with the help of the *Flora of Pakistan* [22,23,24,25]. The specimens were assigned voucher numbers and incorporated into herbaria, subsequently submitted to the Department of Botany, University of Swat, Khyber Pakhtunkhwa, Pakistan. The scientific nomenclature of each taxon was verified through The Plant List database [26] and family assignments were consistent with the Angiosperm Phylogeny Website [27].

### 2.3. Data Analysis

We compared the data with the existing ethnobotanical studies carried out in the Hindukush and Pamir mountain ranges to crosscheck the quoted food uses of the WFPs. The literature survey aimed to compare the current data with previous food ethnobotanical data to identify the novel aspects of this study. On the Pakistani side, we mainly relied on own research work [8,9,10], while for the Afghan Pamir we found only a single study which recorded the ethnobotanies of the Kyrgyz and Wakhi peoples [11]. Similarly, we also compared the data with our research conducted in Tajikistan in the past year [15]. In addition, we considered other medicinal ethnobotanical studies [12,13,14,16] in order to have a comprehensive comparative analysis to establish the novelty of our results. To assess the impact of the lingua franca, i.e., the Wakhi language on the local plant nomenclature of the Sarikoli, we compared the recorded local plant nomenclature among the two selected groups. We also extracted Wakhi plant names from previous ethnobotanical studies [8,11] in order to have a better understanding of the impact of the dominat Wakhi nomenclature on Sarikoli nomenclature and identify the sociolinguistic assimilations of the studied groups in the area, as well as to present sound interpretation of the recorded results.

To determine the relative importance of the recorded taxa, we have also calculated the relative frequency of citation (RFC) using the following formula:RFC = FC/N × 100
where FC indicates the number of informants quoted for a given taxon and N represents the total number of informants.

## 3. Results

### 3.1. WFPs and Their Uses

The current study, which is the first ethnobotanical study in the Wakhan region, has systematically described the traditional uses of WFPs among the Sarikoli and Wakhi. A total of 29 WFP taxa which have been used by the local communities to supplement their local food systems were recorded (Table 2).

The results revealed that the recorded plants were reported by more than fifty percent of the informants from both of the researched groups. Quantitative analysis showed that the local inhabitants quoted comparatively fewer WFP taxa than other local communities from North and West Pakistan (Table 3) whose knowledge of wild resources we previously documented [8,9,10]. This may be due to the fact that these communities have historically been pastoralists, whose traditional food system mainly consists of meat and dairy products, ingredients that are considered very useful as they provide energy during the cold season.

Moreover, they also practice horticulture and grow barley and wild beans, which are highly preferred food items obtained from their local fields. Local inhabitants also import rice from Chitral city, which they frequently use in different seasons throughout the year. We recorded the frequent use of WFPs as cooked vegetables and raw snacks. Very few WFPs were used in teas, lacto-fermentation, or for seasoning. Wild food plants were also frequently used as cooked vegetables and raw snacks among other communities across North and West Pakistan [8,9,10,28,29]. Research has shown that the consumption of raw snacks primarily emerged during the adoption of mobile pastoralism by the studied communities [30]. In Broghil Valley, seasonal movement to grazing pastures is practiced at different times throughout the year. The WFPs recorded during the survey were mainly gathered from crop fields, pastures, and mountains, indicating the prevalence of mixed agropastoral practices, which have shaped the wild food ethnobotanies of the studied groups. The aerial parts of plants were frequently used, and wild vegetables were primarily harvested at young stages of growth. Both male and female community members were involved in gathering WFPs. The consumption of WFPs has decreased in recent times due to social changes and easier access to the local food market. Some of the important WFPs which were reported with a high relative frequency of citation included *Allium carolinianum* (0.90 RFC), *Allium* spp. (0.88), *Berberis calliobotrys* (0.88), *Carum carvi* (0.95), *Eremurus stenophyllus* (0.95), *Mentha longifolia* (1.00), *Rheum ribes* (0.90), and *Ziziphora clinopoides* (0.86). The quoted taxa have also been frequently reported among other remote mountain communities in the Hindukush and other parts of the western belt of Pakistan [8,9,10,11,28,29].

The cross-cultural comparison of WFP uses between the two studied groups revealed a complete homogenization of their wild food ethnobotanies. We argue that the Sarikoli diaspora, who came to the area by the middle of the 20th century, has completely acculturated to Wakhi culture, adopting their traditional lifestyles and customs, which in turn has had irreversible impacts on their local food system and related WFP-centered gastronomic knowledge. To support our argument, we also made a comparison of the quoted local plant names with other ethnobotanical studies (referenced in Section 2.3). We did not find any field ethnobotanical studies conducted among the Sarikoli, but we did find a few studies that were carried out among the Wakhi in the Pamir and Hindukush mountains. The objective was to compare adopted local plant nomenclature in the study area with previous studies, if any, conducted among the studied cultural groups. This comparison was used as a proxy to determine the possible cultural assimilation of the two linguistic groups. We found that all the adopted local names for the quoted taxa were part of Wakhi nomenclature as confirmed by the selected ethnobotanical studies in Wakhan and Gilgit-Baltistan. It is evident that, in a plural society, minority groups always tend to follow the dominant cultural group, and in light of these findings, we assert that the Sarikoli have gone through crucial socio-cultural negotiations, intermarrying with Wakhi and Kho speaking neighbors in Chitral as well as in Gilgit-Baltistan, which in turn has impacted the core body of TEK and hindered its inheritance among the younger generation of Sarikolis. It is interesting to note that Sarikoli people literally identify themselves as “*Sarikoli*”, even though they have completely acculturated to Wakhi culture and have lost their language as well. Qualitative ethnographic observations have confirmed that the local inhabitants are not much involved in gathering WFPs and most of the WFP-centered TEK has become a part of their past.

### 3.2. Comparison with Neighboring Regions

A literature review revealed very few studies on food ethnobotany conducted in the adjoining areas of Pakistan, Afghanistan, China, and Tajikistan (referenced in the Materials and Methods section under Data analysis). After comparing the data with the existing literature, we found some wild food plant ingredients which, to the best of our knowledge, are new to the food ethnobotanical literature of the aforementioned regions: *Zygophyllum obliquum* which is used as a cooked vegetable, *Oxyria*
*digyna* which is especially used for lacto-fermentation but also consumed as a raw snack, *Elaeagnus rhamnoides* whose leaves are used in teas, and *Papaver involucratum* whose flowers are also used in teas.

Since the study area is part of the Hindukush region, we also tabulated (Table 3) the recorded data along with the previously published data from Pakistani Hindukush. Comparison indicated that some of the plants were very rarely reported from the Chitral region, such as *Brassica rapa, Elaeagnus rhamnoides, Lepyrodiclis holosteoides,*
*Malva neglecta*, and *Polygonum* spp. Some of the most common vegetables that are frequently collected in the Northern belt of Pakistan include *Allium carolinianum*, *Eremurus stenophyllus*, *Portulaca* spp., *Chenopodium album*, and *Amaranthus* spp.

### 3.3. Political Boundaries: Threats to Cultural Resilience

Ethnographic observations revealed that third generation Sarikoli migrants have adapted to Wakhi culture, and they speak the dominant Wakhi language, or the Khowar language, as their *lingua franca*. This linguistic adaptation has ultimately come at the expense of the survival of the Sarikoli language, a phenomenon that Skutnabb-Kangas [31] defined as “*linguicide*”.

The results showed that the Sarikoli people have completely adopted the local plant nomenclature of the Wakhi, which demonstrates the significant impact of the *lingua franca* on the diaspora as well as their cultural adaptation to the majority group. Sarikoli participants confirmed that their ancestors, who were traditional healers by profession, fled Taxkorgan Tajik Autonomous County after the arrival of the Communist political system, leaving their relatives behind, and subsequently settled in Broghil Valley by the middle of the 20th century, preferring to live under the political system in that area. Since the political border prevented frequent reunions with relatives, the physical isolation made it difficult for Sarikolis to maintain kinship relations, making them anxious to nurture new relationships. As a result, the Sarikoli entered into marriages with other cultural groups, thus becoming socio-culturally homogenized. However, they still visit their relatives in Tarkushgan as kinship cannot be stopped by artificial (i.e., political) borders. It is worth mentioning that local communities are dealing with different political contexts, and thus they are (willingly or unwillingly) gradually adopting new strategies to cope with the current social and economic challenges, usually engaging in exotic and cross-cultural interactions [32], which affect cultural distinctiveness and threaten future cultural resilience. As resilience is closely linked to the sustainability of cultures, societies, and their respective environments, political pressure on both sides of the border could have negative impacts on cultural survival in the near future. According to Berkes and Ross [33], communities do not control all of the conditions that affect them, but they can change some of the conditions that can increase their resilience. Therefore, it is advisable for state authorities to relax cross-border restrictions, as these communities are politically more vulnerable, socially depressed, and unvoiced. Sometimes, they make small trades across the border as these people are more engaged across the border than within the border, and thus special permits should be allocated to poor local communities in order to obtain their remittances. Flexibility in cross-border movement is an important element in celebrating the biocultural diversity within the whole Pamir region. The Pamir region is a hub of biocultural diversity, and it is crucial to give it the chance to flourish and survive by helping local communities instead of pushing them into difficulty. In the race to globalize and modernize, we have already seen remarkable social change, and the vulnerability of mountain communities is no longer an exception. In fact, unhealthy socio-political practices are leading the planet towards an “extinction crisis*”*, where cultural diversity, linguistic diversity, and biodiversity are equally threatened for one reason or another [34]. It has been estimated that if the current speed of erosion of languages continues, half of all local languages spoken today will disappear by the end of this century see [35]. Similarly, we are rapidly losing biodiversity, and we see incredible homogenization of cultures. As these three domains of life on earth are interconnected, we need to envision future sustainability through a holistic approach. In this regard, we suggest that policy makers focus on cultural diversity and associated TEK in order to foster concrete development tools to help make the planet a better place to sustain life.

### 3.4. Provision of Subsidies to Local Communities: Promotion of Ecotourism

Socio-political marginalization has made the local inhabitants feel underprivileged, and they view themselves as less important because state authorities are constantly undermining the provision of their basic rights and access to local natural resources. On the one hand, these people are exposed to hard political settings, but on the other hand their poor livelihoods and economic instabilities make their lives miserable. Due to the harsh climatic conditions, throughout most of the year, they are not able to grow crops or move freely and they face serious challenges in grazing their animals in pastures. The people are living under poor economic conditions and earn their livelihoods from local subsistence economies. Due to the harsh weather and lack of opportunities, people, especially younger individuals, tend to move to cities in search of better opportunities, and some families have already left Broghil and migrated to nearby valleys. During an informal conversation, when we asked the participants about their life in Broghil, one of the local inhabitants among the Sarikoli stated: “*you should give us a piece of land and we will move there. We know that there is nothing in these mountains, but we can’t move to other areas, we have no alternative solution. We have no jobs, and the government is not helping us*”. In this exceeding challenging situation, the state needs to address local issues, discuss their problems, and grant them special subsidies on local productions to help them improve their subsistence economies. The government should help local communities to produce high quality breeds and manage animal fodder. The government should focus on animal husbandry to improve breeds in order to obtain high quality organic meat. For instance, yak meat is highly valued, and its production could bring economic benefits to the local people. One way to eliminate their economic marginalization is to highlight and promote their food-centered biocultural heritage and the attached TEK, which could open new economic avenues in the study area. It is also important to note that the valley is home to several important medicinal plants, and therefore it would be wise to streamline their collection and value chain to benefit all relevant stakeholders including local communities, and this would also be advantageous for revitalizing local knowledge on medicinal plants [36,37,38]. As affirmed by the 2030 Agenda for Sustainable Development [39] (UN, 2015), culture emerges as a transversal driver, both as knowledge capital and a source of creativity and innovation, as well as a resource to face challenges and find appropriate solutions. Hosagrahar et al. [40] argue that culture should be at the core of developmental policies which are essential for equitable and inclusive development. The area could also be a hot spot of food ecotourism, and so it is shocking to learn that some community members have started coming down from their villages and settling in Mastuj. Therefore, to curb mass migration, there is an urgent need to open new economic avenues for mountain people. In addition, it is relevant to mention that the growth of legal restrictions is also creating several challenges for these agropastoralist societies; for instance, Broghil has been given the status of a national park and the local people are facing problems in grazing their animals. Another informant (a 53-year-old man) among the Sarikoli mentioned his view: “*I do not like the staff of Broghil National Park. If you were a member of staff of Broghil National Park, I would not give you an interview. The staff of the national park behave in an unjust way. We have been banned from grazing our animals, but we do so anyway, as we have no alternatives to feed our animals. As far as the collection of WFPs is concerned, now we don’t do it as we have sufficient food from different sources, i.e., from the market, from fields, as well as the animals that we keep at our houses*” Therefore, political changes, such as collectivization and central planning, are having an impact on long-established patterns of local landscape management. The authorities should respect the rights of the local people and there should be an equitable sharing of benefits from natural resources as the local people are not content with these legal restrictions. The Council of Europe Framework Convention on the Value of Cultural Heritage for Society [41] marked a revolution in the meaning of cultural heritage, shifting attention from objects and places to people. This new way of looking at heritage has laid the foundation for redesigning relationships between all interested stakeholders. During the field survey, some local community members refused to share their knowledge as they considered us staff members of Broghil National Park. As endorsed by the 2030 Agenda for Sustainable development [39], under Sustainable Development Goal 15.4, the conservation of mountain ecosystems, including their biodiversity, should be ensured to enhance their capacity to provide benefits that are essential for sustainable development. However, the agenda does not explicitly stress the needs and priorities of local communities which we think should be emphasized in international development and policy programs. In light of these recommendations, relevant policy makers should ensure the participation of local communities in building developmental and legal infrastructure in the area. It is advisable to develop a system which incorporates and integrates TEK in formulating future developmental strategies in these mountain areas, and this combined approach could support adaptation planning and better respond to community needs.

## 4. Conclusions and Future Recommendations

The current study is an important addition to the food ethnobotany of mountain regions as it reports WFP uses among two highly marginalized linguistic groups in the high mountains of Asia. The results showed remarkable homogenization of WFP uses among the two groups, indicating that Sarikolis have culturally assimilated to the dominant Wakhi culture, which has come at the expense of the survival of the Sarikoli language, as third-generation migrants have completely lost the language. It is worth mentioning that the local plant nomenclature has been completely adopted by the Sarikoli people. The foraging of WFPs and traditional resource management are highly threatened due to various social and political factors. The findings of the current study present useful baseline data to understand the historical socio-cultural adaptations of the studied groups, providing important policy implications for fostering cultural resilience across the region. We appeal to policy makers to address the problems of local communities by enabling them to celebrate their biocultural heritage in future development programs. Traditional ecological knowledge-centered food tourism should be promoted as this will open new avenues of economic development in the study area.

Ethnobiologists should focus on future ethnographic studies in this highly remote mountain region to fully understand the local ecological practices and the possible threats undermining TEK in the region. We especially suggest that TEK be incorporated into future development plans as well as given space in educational platforms to prevent its further erosion as well as the depletion of biocultural diversity.

## Figures and Tables

**Figure 1 biology-11-01543-f001:**
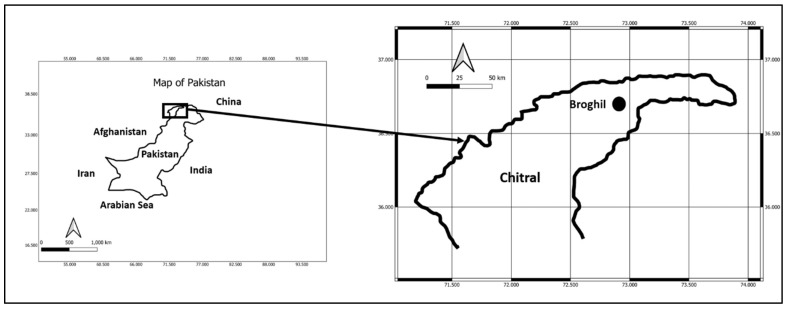
Map of the study area.

**Figure 2 biology-11-01543-f002:**
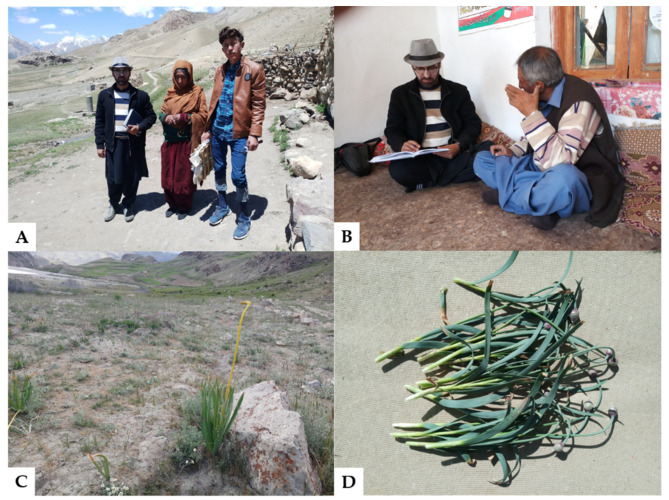
Informants interviewed among the Sarikoli (**A**,**B**); Photos of *Eremurus stenophyllus* (**C**) and *Allium carolinianum* (**D**).

**Table 1 biology-11-01543-t001:** Characteritics of the studied communities.

Langauge	Village	Elevation (m.a.s.l)	Appoximate Number of Individuals	Number of Interviews	Islamic Faith	Subsistance Activites
Wakhi	Lashkargaz	3658	2000	15	Ismaili	Pastoralism/small scale horiticultural practices
Sarikoli	Lashkargaz	3658	60	15	Ismaili	Pastoralism/small scale horiticultural practices

**Table 2 biology-11-01543-t002:** Wild food plants gathered among the two studied linguistic groups.

Botanical Taxon; Family; Botanical Voucher; Specimen Code	Recorded Local Name	Parts Used	Recorded Local Food Uses	Sarikoli	Wakhi	Relative Frequency of Citation	Traditional Food Uses Previously Reported Among the Wakhi in North Pakistan [8]	Previously Reported Taxa among the Wakhi with Similar Local Names [8]
*Allium carolinianum* DC.;Amaryllidaceae; SWAT005988, SWAT005976, SWAT000777	Lanturk ^S, W^	Aerial parts	Cooked ^S, W^	++	++	0.90	Yes	Yes
*Allium* spp.; Amaryllidaceae;	Kach ^S, W^	Aerial parts	Cooked ^S, W^ Salads ^S, W^	++	++	0.88	Yes	Yes
*Amaranthus cruentus* L.; Amaranthaceae; SWAT005512	Sakarghaz ^S, W^	Aerial parts	Cooked ^S, W^	++	++	0.65	Yes	Yes
*Berberis calliobotrys* Bien. ex Koehne; Berberidaceae;SWAT000723	Zolg ^S, W^	Fruits	Raw snacks ^S, W^	++	++	0.88	Yes	Yes
*Brassica rapa* L.; Brassicaceae; SWAT005807, SWAT005520	Chirogh ^S, W^	Leaves	Cooked ^S, W^	++	++	0.55	Yes	Yes
*Carum carvi* L.; Apiaceae; SWAT005486	Nirthak ^S, W^	Seeds	Seasoning ^S, W^ Tea ^S, W^	++	++	0.95	Yes	Yes
*Chenopodium album* L.; Amaranthaceae; SWAT005509, SWAT005499	Shileet ^S, W^	Leaves	Cooked ^S, W^	++	++	0.68		
*Cotoneaster nummularius* Fish. and Mey.; Rosaceae; SWAT005485	Dindlak ^S, W^	Fruits	Raw snacks ^S, W^	++	++	0.52	Yes	Yes
*Elaeagnus rhamnoides* (L.) A.Nelson; Elaeagnaceae: SWAT005998	Khoz gak ^S, W^/Zakh ^S, W^	Fruits, leaves	Fruit: raw snacks ^S, W^, Leaves: tea ^S, W^	++	++	0.53	Yes	Yes
*Eremurus stenophyllus* (Boiss. & Buhse) Baker; Xanthorrhoeaceae; SWAT005967	Laq ^S, W^	Leaves	Cooked ^S, W^	++	++	0.95	Yes	Yes
*Lepyrodiclis holosteoides* (C.A. Mey.)Fenzl ex Fisch. and C.A. Mey.;Caryophyllaceae; SWAT000747	Yarkwush ^S, W^	Aerial parts	Cooked ^S, W^	++	++	0.54	Yes	Yes
*Malva neglecta* Wallr.; Malvaceae; SWAT006043	Swachal ^S, W^	Leaves	Cooked ^S, W^	++	++	0.56	Yes	Yes
*Medicago sativa* L.; Fabaceae; SWAT005797, SWAT005795	Wujark ^S, W^	Aerial parts	Cooked ^S, W^	++	++	0.58	Yes	Yes
*Mentha longifolia* (L.) L.; Lamiaceae; SWAT005792, SWAT005790	Wadin ^S, W^	Aerial parts	Salads ^S, W^	++	++	1.00	Yes	Yes
*Oxyria digyna* (L.) Hill.; Polygonaceae; SWAT006053	Trush pop ^S, W^	Leaves	Raw snacks ^S, W^ Lacto-fermentation: milk into yoghurt ^S, W^	++	++	0.78	Yes	Yes
*Papaver involucratum* Popov.; Papaveraceae; SWAT000744	Gulmarwai ^S, W^	Flowers	Tea ^S, W^	++	++	0.50	Yes	Yes
*Polygonum* spp.; Polygonaceae; SWAT006051	Wingasgas ^S, W^/Wingsgos ^S, W^	Leaves	Leaves: raw snacks ^S, W^	++	++	0.52	Yes	Yes
*Rheum ribes* L.; Polygonaceae; SWAT004749	Ishpat ^S, W^	Young stems	Raw snacks ^S, W^ Lacto-fermentation: milk into yogurt ^S, W^	++	++	0.90	Yes	Yes
*Ribes alpestre* Wall.ex. Decne.; Grossulariaceae; SWAT005775	Chilazum ^S, W^	Fruits	Raw snacks ^S, W^	++	++	0.82	Yes	Yes
*Ribes orientale* Desf.; Grossulariaceae; SWAT005774, SWAT005971	Ginat ^S, W^	Fruits	Raw snacks ^S, W^	++	++	0.84	Yes	Yes
*Rosa webbiana* Wall. ex Royle; Rosaceae; SWAT005502	Charir ^S, W^	Fruits, leaves	Fruit: raw snacks ^S, W^ Leaves: tea ^S, W^	++	++	0.62	Yes	Yes
*Rumex dentatus* L.; Polygonaceae; SWAT005468	Shalkha ^S, W^	Leaves	Cooked ^S, W^	++	++	0.78	Yes	Yes
*Taraxacum campylodes* G.E.Haglund;Asteraceae; SWAT005972	Paps/Papas ^S, W^	Leaves	Cooked ^S, W^	++	++	0.68	Yes	Yes
*Tulipa* spp.; SWAT006052	Sai shulam ^S, W^		Raw snacks ^S, W^	++	++	0.50	Yes	Yes
*Zygophyllum obliquum* Popov. Zygophyllaceae; SWAT006049	Yum wush ^S, W^	Aerial parts	Cooked ^S, W^	++	++	0.58	Yes	Yes
*Ziziphora clinopoides* Lam. Lamiaceae; SWAT006050	Jambilak ^S, W^	Leaves	Tea ^S, W^	++	++	0.86	Yes	Yes
Unidentified taxon	Jarjwush ^S, W^	Aerial parts	Raw snacks ^S, W^	++	++	0.50	Yes	Yes

^S^: Local plant name recorded among the Sarikoli people; ^W^: Local plant name recorded among the Wakhi people. ++: food use quoted by more than 50% of the study participants.

**Table 3 biology-11-01543-t003:** Presence-absence index for the plants reported among the seven researched groups in Chitral [9,10].

Plant Taxa	Sarikoli	Kho	Kalasha	Kamkatawari	Wakhi	Yidgha
*Alluim* spp.	+	+	+	+	+	+
*Amaranthus* spp.	+	+	+	+	+	+
*Berberis lycium*	+	+	+	+	+	+
*Brassica rapa*	+	-	-	-	+	-
*Carum carvi*	+	+	+	+	+	+
*Chenopodium album*	+	+	+	+	+	+
*Cotoneaster nummularius*	+	+	+	+	+	+
*Elaeagnus rhamnoides*	+	+	-	-	-	-
*Eremurus* spp.	+	+	+	+	+	+
*Lepyrodiclis holosteoides*	**+**	**+**	**-**	**-**	**-**	**-**
*Malva neglecta*	+	+	-	-	-	-
*Medicago sativa*	+	+	+	+	-	+
*Mentha longifolia*	+	+	+	+	+	+
*Oxyria digyna*	+	-	-	-	-	-
*Papaver involucratum*	+	-	-	-	-	-
*Polygonum* spp.	+	-	-	-	+	-
*Portulaca* spp.	+	+	+	+	+	+
*Rheum* spp.	+	+	+	+	+	+
*Ribes alpestre*	+	-	-	+	-	+
*Ribes orientale*	+	-	-	+	-	+
*Rosa webbiana*	+	+	-	-	-	-
*Rumex* spp.	+	+	+	+	+	+
*Taraxacum campylodes*	+	+	+	+	+	+
*Tulipa* spp.	+	+	+	+	-	+
*Ziziphora clinopoides*	+	-	-	+	+	+
*Zygophyllum obliquum*	+	-	-	-	-	-

+: presence; -:absence.

## Data Availability

All the data is available in this article.

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
