# Peer review of "Plant Use Adaptation in Pamir: Sarikoli Foraging in the Wakhan Area, Northern Pakistan"

_biology, 2022, doi:10.3390/biology11101543_

Round 1

Reviewer 1 Report

Kindly see my comments attached 

Author Response

Reviewer 1 comments:

  1. Please check the list of authors

CORRECTED.

  1. Please check the text to avoid self-plagiarism, you have another preprint published and there is an important similarity

THE PRE-PRINT HAS BEEN REMOVED WHICH WAS PUBLISHED BY ANOTHER JOURNAL FOR THE PREVIOUSLY SUBMITTED VERSION OF THE MS.

  1. Please explain the motivation of preparing this work

WE ADDED A PHRASE ON THE MOTIVATION OF THIS WORK IN THE INTRODUCTION SECTION. SEE THE HIGHLIGHTED TEXT.

  1. Please provide more details (description of the sample and main variables you considered in the present work).

WE REVISED THE METHODOLOGICAL SECTION.

  1. Can't understand the employed methodology? it would be great to have more details about the methodology and to explain your contribution to the literature.

WE REVISED THE METHODOLOGICAL SECTION.

  1. You can add a comparison with other countries (developing and developed countries)

SEE SECTION 3.2.

  1. It would be great to include more references from MDPI journals dealing with similar topics for instance :

1- Taghouti, I.; Cristobal, R.; Brenko, A.; Stara, K.; Markos, N.; Chapelet, B.; Hamrouni, L.; Buršić, D.; Bonet, J.-A. The Market Evolution of Medicinal and Aromatic Plants: A Global Supply Chain Analysis and an Application of the Delphi Method in the Mediterranean Area. Forests 2022, 13, 808. https://doi.org/10.3390/f13050808

2- TAGHOUTI, Ibtissem; DALY-HASSEN, Hamed. Essential oils value chain in Tunisian forests: Conflicts between inclusiveness and marketing performance. Arabian Journal of Medicinal and Aromatic Plants, [S.l.], v. 4, n. 2, p. 15-41, nov. 2018. ISSN 2458-5920. Available at: <https://revues.imist.ma/index.php/AJMAP/article/view/1424>. Date accessed: 23 sep. 2022. doi:https://doi.org/10.48347/IMIST.PRSM/ajmap-v4i2.14249.

3- Taghouti, I.; Ouertani, E.; Guesmi, B. The Contribution of Non-Wood Forest Products to Rural Livelihoods in Tunisia: The Case of Aleppo Pine. Forests 2021, 12, 1793. https://doi.org/10.3390/f12121793.

WE ADDED THE RECOMMENDED CITATIONS.

Reviewer 2 Report

This paper analyses the wild plant foraging practices of ethnic groups in Pakistan, the Sarikoli and Wakhi. It aims to understand the role of wild food plants in the cultural and historical contexts of the communities. While this is a very interesting paper, the authors should revise some issues:

- the authors should expand the analysis of previous research about wild food plants in the introduction and the use of references to describe the study area in section 2.1.

- the authors should provide the profile of the interviewees in the method.

- the authors should expand the results with direct quotes from the interviews and pictures.

- the authors should expand theoretical and practical implications, limitations and opportunities for further research.

Author Response

  1. The authors should expand the analysis of previous research about wild food plants in the introduction and the use of references to describe the study area in section 2.1.

WE HAVE ADDED NEW TEXT TO THE INTRODUCTION ALONG WITH PROPER CITATIONS. ALSO, WE HAVE ADDED THE RELEVANT REFERENCES FROM THE INTRODUCTION SECTION TO THE METHODOLOGY SECTION.

  1. The authors should provide the profile of the interviewees in the method.

A TABLE ON THE CHARACTERISTICS OF THE STUDIED COMMUNITIES HAS BEEN ADDED.

  1. The authors should expand the results with direct quotes from the interviews and pictures.

DONE.

  1. The authors should expand theoretical and practical implications, limitations and opportunities for further research.

NEW TEXT WAS ADDED AND HIGHLIGHTED IN THE DISCUSSION AND CONCLUSION SECTIONS.

Reviewer 3 Report

The manuscript titled “One century of food adaptation: Sarikoli wild food plant uses in the Wakhan Area, Northern Pakistan” describes the cross-cultural ethnobotanical study, the food uses of wild food plants (WFPs) were recorded among two linguistic groups. The authors have discussed the local knowledge linkages to wild food plants to particular locality in Pakistan close to Afghanistan border. They have highlighted that local knowledge of WFPs is quite threatened in the two considered communities due to certain socio-political factors.

 Over all, I appreciate the authors for such a valuable effort. However, my concerns are the lack of proper research design in methodology, mostly a descriptive work, interpretation of results and quality of presentation. As it is submitted to a high impact factor journal “Biology”, we need to check how important it is for the wider audiences. I believe that it will attract only specific/regional audiences/readers and will not be of significant importance to the broader community.

I also couldn’t find any ethical approval from a suitable Institutional Review Board or organization. The study involves information related to human culture and ethnobotanic uses involving questionnaire data collection.

I would like to highlight issues in different sections of the manuscript.

Title: Needs improvement e.g. Should be supported by literature. In the text the authors mentioned that some individual told them their parents moved into this area a century ago. I believe that the authors have based the title on that single statement.

Authors list: Is the authors list complete? Why is there an “and” at the end?

Abstract

- Methodology part is missing apart from the mention of questionnaire.

- Results part is very descriptive and exaggerated, needs revision

- Conclusion missing, no link established with the results

Introduction

The introduction has included some good literature, but more than 50% of citations are older than 5 years. Some technical issues are highlighted below.

- Line 61. 3000 m a.s.l expand this unit.

- Line 69, 70. 50 to 60? Population 18,000 (write population)

- Line 80 and 81. Sentence needs revision.

- Line 85, (a), parenthesis missing.

Material and methods

The methodology is very clear, apart from a questionnaire data, nothing else is significant. Proper experimental design is missing. Any solid statistical methods applied to the analyses of the questionnaire data? Any statistical comparison done with the point in Line 152?

-Line 97, 98, 109. Add a.s.l to all values (consistency)

- Line 100. Area is declared as national park? Any authority?

- Line 105. Sentence needs revision.

-Line 112 to 122. Reference is anecdotal. Needs suitable authority.

-Line 152, We compare the data with the existing ethnobotany studies….  How did you compare the data? What method did you use? what was the purpose of this comparison? Any results obtained presented in the results section?

- Proper hypothesis needed

Results

The results are very descriptive, apart from a single table of records, no concrete analyses presented. Other issues are:

-Line 160 to 170, background information, not results from the study.

-Line 177. Quantitative analysis showed….Where is the quantity? I could only see descriptions.

-Line 189. References Aziz et al. 2020 a, 2020b 2021 a b. Check properly for accuracy. Refere to References section too.

Line 256 to 265. Interesting thinking, Is this part of the results obtained? Or is it the summary of findings? Or some subjective description?

Line 313. Point 38, elaborate what point? Is it an article?

Conclusion

-Please focus on the main findings of the work and link your conclusions to your actual findings.

Figures

Figure 1. Area map is very basic and blurry, needs improvements. Coordinates missing, global/wider location missing?

Figure 2. Multiple images, needs description of each in the caption and proper labelling.

Table

Caption of the table is missing, descriptive key to symbols used is missing.

Formatting need assessment according to journal

References

Check Reference 2, 3 4, and 6 for accuracy.

Author Response

  1. I also couldn’t find any ethical approval from a suitable Institutional Review Board or organization. The study involves information related to human culture and ethnobotanic uses involving questionnaire data collection.

DETAILS HAVE BEEN PROVIDED IN THE METHODS SECTION.

  1. Title: Needs improvement e.g. Should be supported by literature. In the text the authors mentioned that some individual told them their parents moved into this area a century ago. I believe that the authors have based the title on that single statement.

CHANGED.

  1. Authors list: Is the authors list complete? Why is there an “and” at the end?

CORRECTED.

  1. In Abstract:
  2. a) - Methodology part is missing apart from the mention of questionnaire.
  3. b) - Results part is very descriptive and exaggerated, needs revision

c)- Conclusion missing, no link established with the results

WE REVISED AND ADDED THE REQUIRED INFORMATION IN THE RELEVANT PARTS OF THE ABSTRACT.

  1. Introduction: The introduction has included some good literature, but more than 50% of citations are older than 5 years. Some technical issues are highlighted below.

THE INTRODUCTION HAS ALSO BEEN REVISED AND NEW LITERATURE HAS BEEN CITED.

  1. - Line 61. 3000 m a.s.l expand this unit; Line 69, 70. 50 to 60? Population 18,000 (write population); Line 80 and 81. Sentence needs revision; Line 85, (a), parenthesis missing.

CORRECTED.

  1. Material and methods: The methodology is very clear, apart from a questionnaire data, nothing else is significant. Proper experimental design is missing. Any solid statistical methods applied to the analyses of the questionnaire data? Any statistical comparison done with the point in Line 152?

THE SECTION HAS BEEN PROPERLY REVIEWED AND REVISED BY ADDING THE IMPORTANT INFORMATION.

  1. -Line 97, 98, 109. Add a.s.l to all values (consistency)

CORRECTED.

  1. Line 100. Area is declared as national park? Any authority?-

WE ADDED A REFERENCE.

  1. Line 105. Sentence needs revision.

DONE.

  1. Line 112 to 122. Reference is anecdotal. Needs suitable authority.

WE ADDED AN AUTHORITY.

  1. -Line 152: We compare the data with the existing ethnobotany studies….  How did you compare the data? What method did you use? what was the purpose of this comparison? Any results obtained presented in the results section? Proper hypothesis needed.

THE WHOLE SECTION ON DATA ANALYSIS HAS BEEN REVISED AND IMPROVED BY SUPPLEMENTING THE REQUIRED INFORMATION.

  1. Results: The results are very descriptive, apart from a single table of records, no concrete analyses presented.

A NEW SECTION (3.2) HAS BEEN PROVIDED WHICH PRESENTS A COMPREHENSIVE OVERVIEW OF THE ANALYTICAL PART OF THE ARTICLE.

  1. Other issues are:-Line 160 to 170, background information, not results from the study.

WE REMOVED THE TEXT.

  1. Line 177. Quantitative analysis showed….Where is the quantity? I could only see descriptions.

WE REVISED THE STATEMENT.

  1. -Line 189. References Aziz et al. 2020 a, 2020b 2021 a b. Check properly for accuracy. References section too.

WE CORRECTED THE CITATIONS AS WELL AS THE REFERENCES.

  1. Line 256 to 265. Interesting thinking, Is this part of the results obtained? Or is it the summary of findings? Or some subjective description?

IT IS A SUBJECTIVE DESCRIPTION WHICH WE THOUGHT WOULD BE BETTER PLACED HERE.

  1. Line 313. Point 38, elaborate what point? Is it an article?

EXPLAINED.

  1. Conclusion: Please focus on the main findings of the work and link your conclusions to your actual findings.

WE REVISED THE WHOLE SECTION.

  1. Figures: Figure 1. Area map is very basic and blurry, needs improvements. Coordinates missing, global/wider location missing?

A NEW MAP HAS BEEN PROVIDED.

  1. Figure 2. Multiple images, needs description of each in the caption and proper labelling.

DONE.

  1. Table: Caption of the table is missing, descriptive key to symbols used is missing. Formatting need assessment according to journal.

DONE.

  1. References: Check Reference 2, 3 4, and 6 for accuracy.

WE CORRECTED AND FORMATTED ACCORDING TO THE JOURNAL STYLE.

Round 2

Reviewer 1 Report

The manuscript has improved substantially

Please try to improve table 1

Author Response

Reviewer 1 comment:

  1. Please try to improve table 1

DONE.

Reviewer 2 Report

The authors have revised the paper.

Author Response

DONE. 

Reviewer 3 Report

The manuscript titled “Plant use adaptation in Pamir: Sarikoli foraging in the Wakhan 2 Area, Northern Pakistan” describes the cross-cultural ethnobotanical study, the food uses of wild food plants (WFPs) were recorded among two linguistic groups. The authors have discussed the local knowledge linkages to wild food plants to particular locality in Pakistan close to Afghanistan border.

However, I still have serious concerns about the research design in methodology and interpretation of results.

I would like to highlight issues in different sections of the manuscript.

Title: This has been modified according to the recommendations, the word “Pamir” used in the title, is it a well-known place internationally, if not, please add other indicators to the title to identify this study area.

Simple Summary:

Line 26           quite threatened, can you change it to something appropriate (choice of words).

Line 26-30      Authors have highlighted and put forwarded recommendations related to legal restrictions and border crossing. Is this a key area researched in this article?

Abstract

Line 37-          Methodology part is added now. The authors mentioned that they have only 30 participants in this study. How significant this number is statically for sampling a population of over 2000 individuals?

Line 42-          Reads as” this could be contributed to…”is very descriptive and hypothetical, not based on the actual results.

Line 47-          Reads as “Legal restrictions and sanctions on accessing….” I couldn’t see any analysis related to these parameters in the methodology and results sections.  

Line 52-          erosion in the region (Soil erosion, cultural erosion??)

Introduction

Recent literature added. Significantly improved.

Line 122-128  Objectives are mentioned in these lines. These objectives should be achieved by using suitable methods. For example, what variables were used in the methods section to achieve the “understanding of the historical cultural adaptation process”?

Material and methods

-Line 133-185. Study area description, best suited to the introduction part.

- Line 179-181. Revise sentence.

- Line 105. Sentence needs revision.

Line 231 and 233        Comparative analysis: did you do just visual analyses or did you use any software or tools? Why didn’t you use any statistical indices for your analyses?

My concern is the sampling size. Do you think 30 individuals, 15 each from two communities of over 2000 is a good sampling size. A good sampling size is around 10%, your sampling size is less than 2%? Don’t you think that will have significant impact on your results?

Results

The results are still very descriptive, Table 3 has been added to the results but it is still a descriptive table. Other issues are:

-Line 251                    Quantitative analysis …. (Where is that analyses?) Table 3 is merely presence-absence data. No quantity there.

-Line 177. (Previous comments) Quantitative analysis showed….Where is the quantity? I could only see descriptions.

Section 3.3                  Line 328-350 This part of the manuscript provides a good commentary and historical data about the region but is not directly linked with the results obtained.

Line 351 – 354            Is there any data to support this?

Line 365-371              Reads as” it has been estimated…..” How did you estimate this? Any data to support this?

Line 367                      “basic rights” What basic rights? Have you collected any data regarding these rights? This could be something significant if supported by data.

Line 398                      “the Valley” refereeing to Pamir Valley? Or Chitral?

Conclusion

-        Improved conclusions this time.

Line 456 and 459        Celebrate their biocultural heritage.

Figures

Better figure now. Do you want to mention Pamir somewhere too, as per title?

References

Improved!

Author Response

Reviewer 3 comments:

  1. Title: This has been modified according to the recommendations, the word “Pamir” used in the title, is it a well-known place internationally, if not, please add other indicators to the title to identify this study area.

WE HUMBLY BELIEVE THAT PAMIR IS A WELL-KNOWN MOUNTAIN RANGE IN CENTRAL ASIA. WE THINK EVERYONE DEALING WITH SCIENCE SHOULD KNOW IT.

  1. Simple Summary: Line 26          quitethreatened, can you change it to something appropriate (choice of words).

DONE.

  1. Line 26-30      Authors have highlighted and put forwarded recommendations related to legal restrictions and border crossing. Is this a key area researched in this article?

THESE ARE MINOR FUTURE RECOMMENDATIONS WHICH WE PROPOSED FOR POLICY MAKERS. WE HAVE MADE THEM ON THE BASIS OF OUR GROUND OBSERVATIONS AND INTERVIEW DISCUSSIONS ON THIS SUBJECT. THE DISCUSSION SECTION ADDRESSES THIS POINT AS WELL.

  1. Abstract Line 37- Methodology part is added now. The authors mentioned that they have only 30 participants in this study. How significant this number is statically for sampling a population of over 2000 individuals?

WE HAVE CLARIFIED THIS POINT IN THE MATERIALS AND METHODS SECTION. YOU WERE QUITE RIGHT IN COMMENTING HERE. WE CHECKED THE TABLE 1. WHICH WE HAVE PROVIDED WITH WRONG FIGURES AND DATA FOR THE TWO RESEARCHED GROUPS. IN THE PREVIOUS REVISION WE HAVE REFERRED DATA ON POPULATION FOR WAKHI TO SARIKOLI AND VICE VERSA. MOREOVER WE HAVE ALSO EXPLAINED THE ISSUE ON THE NUMBER OF THE PARTICIPANTS IN THE MM SECTION AS THE ABSTRACT HAS A WORD LIMIT.

  1. Line 42-          Reads as” this could be contributed to…”is very descriptive and hypothetical, not based on the actual results.

THIS HAS NOW BEEN AMENDED.

  1. Line 47-        Reads as “Legal restrictions and sanctions on accessing….” I couldn’t see any analysis related to these parameters in the methodology and results sections.  

THESE WERE IN FACT NOT THE PILLARS OF OUR RESEARCH OBJECTIVES NOT THE INTENDED PARAMETERS TO BE MEASURED IN THE STUDY, BUT SINCE THE STUDY PARTICIPANTS MENTIONED THESE ISSUES WE BELIEVED THAT IT WAS QUITE IMPORTANT TO HIGHLIGHT THEM; MOREOVER THIS SPECIAL ISSUES IS SPECIFICALLY DEVOTED TO BRIDGE THE GAP BETWEEN BIOLOGY AND SOCIAL SCIENCES/HUMANITIES AND THEREFORE WE BELIEVED THESE REFLECTIONS ARE NOT MISPLACED.

.

  1. Line 52-          erosion in the region (Soil erosion, cultural erosion??)

THE EDITOR HAVE NOW CLARIFIED THEM.

  1. Line 122-128  Objectives are mentioned in these lines. These objectives should be achieved by using suitable methods. For example, what variables were used in the methods section to achieve the “understanding of the historical cultural adaptation process”?

WE HUMBLY BELIEVE WE HAVE MADE ENOUGH CLEAR THAT WE WANTED TO CROSS-CULTURALLY COMPARE WFP USES AMONG THESE TWO GROUPS AND UNDERSTAND IF ETHNBOTANICAL DATA SHOW WHETHER SARIKOLI UNDERWENT A CULTURAL ADAPTATION OR NOT.

  1. Material and methods; Line 133-185. Study area description, best suited to the introduction part.

WE HONESTLY THINK IT IS BETTER TO RETAIN THE TEXT IN THE MM SECTION AS ALSO SUGGESTED BY THE OTHER REVIEWERS.

  1. Line 179-181. Revise sentence.

CORRECTED.

  1. Line 105. Sentence needs revision.

CORRECTED.

  1. Line 231 and 233        Comparative analysis: did you do just visual analyses or did you use any software or tools? Why didn’t you use any statistical indices for your analyses?

IN THIS REVISION WE HAVE PRESENTED THE REPORTED DATA ALONG WITH THE RFC.

  1. My concern is the sampling size. Do you think 30 individuals, 15 each from two communities of over 2000 is a good sampling size. A good sampling size is around 10%, your sampling size is less than 2%? Don’t you think that will have significant impact on your results?

WE CLARIFIED THIS ISSUE IN THE MM SECTION. SEE THE TEXT WHICH WE HAVE HIGHLIGHTED IN RED.

  1. Results: The results are still very descriptive, Table 3 has been added to the results but it is still a descriptive table.

THE RESULTS HAVE BEEN DISCUSSED IN THE LIGHT OF THE RFC ANALYSIS.

  1. Line 251                    Quantitative analysis …. (Where is that analyses?) Table 3 is merely presence-absence data. No quantity there.

THE RESULTS HAVE BEEN DISCUSSED IN THE LIGHT OF THE RFC ANALYSIS.

  1. Line 177. (Previous comments) Quantitative analysis showed….Where is the quantity? I could only see descriptions.

THE RESULTS HAVE BEEN DISCUSSED IN THE LIGHT OF THE RFC ANALYSIS.

  1. Section 3.3   Line 328-350 This part of the manuscript provides a good commentary and historical data about the region but is not directly linked with the results obtained.

DONE.

  1. Line 351 – 354            Is there any data to support this?

WE GATHERED DATA THROUGH DIRECT OBSERVATIONS DURING THE STUDY. WE DID NOT FIND ANY LITERATURE SOURCES TO CITE IN RELATION TO THIS.

  1. Line 365-371              Reads as” it has been estimated…..” How did you estimate this? Any data to support this?

WE HAVE ADDED A REFERENCE.

  1. Line 367                      “basic rights” What basic rights? Have you collected any data regarding these rights? This could be something significant if supported by data.

WE GATHERED DATA THROUGH DIRECT OBSERVATIONS DURING THE STUDY. WE DID NOT FIND ANY LITERATURE SOURCES TO CITE IN RELATION TO THIS.

  1. Line 398                      “the Valley” refereeing to Pamir Valley? Or Chitral?

HERE WE MEAN THE STUDY SITE “BROGHIL VALLEY”. WE HAVE ADDED IT TO THE TEXT.

  1. Conclusion; Line 456 and 459Celebrate their biocultural heritage.

WE REMOVED THE REPETITION.

  1. Figures; Better figure now. Do you want to mention Pamir somewhere too, as per title?

PAMIR IS A VERY L-A-R-G-E MOUNTAINOUS RANGE, AND WE CANNOT SPECIFY IT ON THE MAP WE HAVE PROVIDED. WE THINK IT IS BETTER TO LEAVE THE MAP AS WE HAVE IT RIGHT NOW.

Round 3

Reviewer 3 Report

All queries addressed properly.  Plausible explanation to the methodology part. Thank you !